## Article

# Flexible and Efficient Multi-Keyword Ranked Searchable Attribute-Based Encryption Schemes

Je-Kuan Lin [1], Wun-Ting Lin [1,*] and Ja-Ling Wu [1,2,*]

1    Department of Computer Science and Information Engineering, National Taiwan University,
Taipei City 10617, Taiwan
2    Graduate Institute of Networking and Multimedia, National Taiwan University, Taipei City 10617, Taiwan
*    Correspondence: d07922020@ntu.edu.tw (W.-T.L.); wjl@cmlab.csie.ntu.edu.tw (J.-L.W.)

**Abstract:** Currently, cloud computing has become increasingly popular and thus, many people and institutions choose to put their data into the cloud instead of local environments. Given the massive amount of data and the fidelity of cloud servers, adequate security protection and efficient retrieval mechanisms for stored data have become critical problems. Attribute-based encryption brings the ability of fine-grained access control and can achieve a direct encrypted data search while being combined with searchable encryption algorithms. However, most existing schemes only support single-keyword or provide no ranking searching results, which could be inflexible and inefficient in satisfying the real world's actual needs. We propose a flexible multi-keyword ranked searchable attribute-based scheme using search trees to overcome the above-mentioned problems, allowing users to combine their fuzzy searching keywords with AND–OR logic gates. Moreover, our enhanced scheme not only improves its privacy protection but also goes a step further to apply a semantic search to boost the flexibility and the searching experience of users. With the proposed index-table method and the tree-based searching algorithm, we proved the efficiency and security of our schemes through a series of analyses and experiments.

**Keywords:** attribute-based encryption; searchable encryption; index table; fuzzy search; semantic search; data retrieval





## 1. Introduction

### 1.1. Motivations

Cloud and IoT [1] services have become increasingly popular because of the rise in streaming services [2] and the development of machine learning, especially in the era of COVID-19. Outsourcing data to the cloud saves space for local storage and brings convenience so that users can access and share their data without any space and time limitations. However, since cloud service providers, or cloud servers for short, are not fully trustable, directly uploading sensitive data to the cloud is dangerous and undermines user privacy. Encrypting data and then uploading them seems a safer approach. Nevertheless, in many situations, traditional public key encryption (PKE) [3] schemes can only achieve secrecy but lack proper access controllability. For example, in some cases, we want to authorize files to only a specified group of people. Under PKE, we must copy files many times and encrypt them, respectively. Moreover, the management of secret keys is increasingly cumbersome and difficult. This challenge is specifically severe for medical and financial data because users have the right to decide who can review their sensitive medical and financial records. With attribute-based encryption (ABE) [4–9], we can make fine-grained access control much more manageable by only allowing some people with specified attributes (i.e., conditions) to access and view the files.

In addition to the access control, how to fetch the required data rapidly among the massive data stored in the cloud is also a critical issue. Downloading and decrypting all

the data and then performing a search can reach the target, but it is not feasible because a massive amount of computation and storage is required on the user end. Apart from the excessive time overhead, these operations may be unsafe. Searchable encryption (SE) algorithms [10–15] bring reasonable solutions to this problem. Go a step further; combining the ABE and SE schemes allows users to have fine-grained access controls and searching capabilities regarding encrypted data.

Many searchable attribute-based encryption schemes (ABS) [16–24] have provided fine-grained access control, dynamic updates, and attribute revocations. However, searching capabilities could be more potent in most schemes to fulfill actual needs. Usually, they can embed only a single keyword into ciphertexts, which could be inconvenient and make searching more cumbersome. Although some schemes allow for combining multiple keywords and provide ranked search results, users can only fetch files containing all the keywords. More complicated relationships between keywords such as disjunctive logic "OR" can usually not be expressed. In addition, some advanced designs in searchable encryption algorithms have rarely been implemented on such systems. We summarize the standard advanced searching modes in Figure 1. The basic search mode is the keyword rank search which does the exact match of single or multiple keywords. However, in practice, the user's input commonly contains some typos or uses synonyms. As a result, two high-level search modes, fuzzy search and semantic search, are induced to allow users to obtain the results without using the exact keyword.

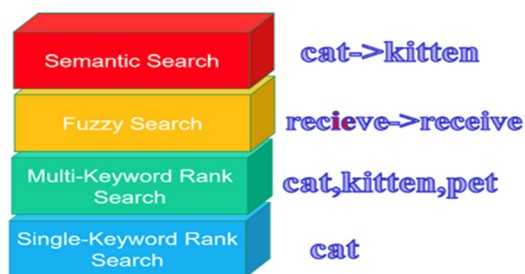

**Figure 1.** The hierarchy of standard searching modes and the concrete associated examples. Unlike single and multi-keyword rank search, fuzzy and semantic search belong to high-level searching modes.

To tackle the problems listed above, we proposed two flexible and efficient multi-keyword ranked searchable attribute-based encryption schemes (FEMRSABE), which are especially suitable for E-health applications. In our basic scheme, we designed a search tree data structure to enhance the expressiveness of the search, as shown in Figure 2. The server matches the trapdoors in leaf nodes with index files, traversing the tree and inducing the searching results of parent nodes by union or intersection. Finally, the aggregated search result of the root node is the final result, sorted according to the associated relevance score. The cloud server can only read the user-inputted logic structure but knows nothing about what users have searched. In addition, inspired by [25], we built an index table to boost search efficiency. We replaced the encryption mechanism from symmetric key encryption with pure attribute-based encryption. Data owners do not need to exchange keys with users in advance, making the scheme more realistic. It shows that the search speed is much faster than the case without the index table through experiments. We also provide fuzzy keyword searching ability by calculating the fingerprints of keywords. We refer to the generating method and the similarity score in [11] to ensure the search range is manageable.

Moreover, in our enhanced scheme, we reorganized the system architecture to minimize possible data leakages, such as the logical structure of search trees and the file list of a particular keyword. We further implemented the semantic search functionality with WordNet's help [26]. As a consequence, we considered the actual semantics of the keywords. Users only need to express their intention of searching without considering the constraints on the data owners' actual keywords and their perfect spellings. These advanced search modes make the search procedure more flexible and easier to use. The functionality com-

parisons in later sections show that our scheme has more desirable searching capabilities than other benchmarking searchable attribute-based encryption schemes.

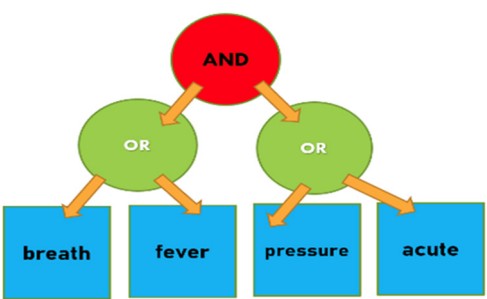

**Figure 2.** This work uses a tree-based data structure and AND–OR gates to complete a complicated keyword search task in the encryption domain. This is an example of an E-health use case.

Flexible and efficient multi-keyword ranked searchable attribute-based encryption schemes (FEMRSABE) target the E-health use case. Users, who are equipped with IoT devices that can collect body data such as heart rate and body temperature, can upload their data to the server in encryption form. Doctors or other healthcare professionals can access the data with appropriate permission. Most importantly, the system does not require the accessor to input the exact identical keyword used for encryption. With the benefit of fuzzy and semantic search, FEMRSABE can automatically match and discover the possible meaning and search. This brings flexibility in that IoT providers and healthcare professionals do not need to negotiate the keyword beforehand, and the different IoT devices' cross-time can also pick up a suitable keyword rather than be limited to the previous choice.

In the security aspect, our FEMRSABE system can defend against selective cipher-text-policy and chosen-plaintext attack (IND-SCP-CPA) by building it on the general bilinear map cryptographic techniques and the associated assumptions.

### 1.2. Contributions of This Work

We proposed a flexible and efficient scheme, FEMRSABE. The possible contributions of this work include the following:

**Flexible Access and Searching Structure:** We used linear secret-sharing schemes (LSSS) to build the basic data access structure, allowing data owners to express their data access policy by combining AND–OR logic gates as their wish. Furthermore, the conventional multi-keyword scheme can only find documents containing all the searching keywords. We designed a much more flexible tree structure so that users can express what they want to search by both conjunctive and disjunctive logic.

**Ranked Searching Results:** Following the techniques presented in [25], we built an index-table structure that can diminish the searching time and make ordered searching results possible. Users can obtain the most desired search results as soon as possible, avoiding unnecessary file decryption or filtering among many matched results.

**Fuzzy and Semantic Search Mode:** We further included advanced search mechanisms into the enhanced scheme, such as fuzzy and semantic search, by integrating with fingerprints introduced by [11]. Query keywords can now be inaccurate or have spelling errors, making it easier for users to obtain what they want.

**Multi-Authority:** Allowing the central authority to take over all the jobs of generating user keys is neither efficient nor secure. If the central authority shuts down, the whole system will be affected, which is called the "single-point" failure. We set up multiple attribute authorities to spread the traffic and generate intermediate user keys to solve this problem and shorten the key-generating time.

### 1.3. Organization

This paper is organized as follows. We review some related attribute-based and searchable encryption schemes in Section 2. Some preliminaries and cryptography backgrounds are addressed in Section 3. Section 4 defines the problem formally and depicts the proposed architecture, while Section 5 addresses our concrete constructions in detail. We present our schemes' performances and security levels through a series of experiments in Section 6. Finally, Section 7 concludes this write-up.

## 2. Related Work

### 2.1. Attribute-Based Encryption

Attribute-based encryption (ABE) is a technique that allows data owners to declare their access policies such as: "*(Doctor OR Researcher) AND (Chest OR Surgery)*". Only data users who meet the policy's attribute requirements are qualified to access the files. For instance, users with the attributes "Doctor and Surgery" can read the text, but ones with "Doctor and Researcher" cannot. Most ABE schemes can be categorized into the following two classes: ciphertext-policy attribute-based encryption (CP-ABE) and key-policy attribute-based encryption (KP-ABE). Wang et al. [27] proposed a constant-size ciphertext KP-ABE scheme, while Water et al. [4] proposed the first practical CP-ABE scheme. The main difference between KP-ABE and CP-ABE is that CP-ABE puts the access policy into ciphertexts while KP-ABE puts it into the users' secret keys. In CP-ABE schemes, data owners can easily decide who can access the files, so it is more suitable for cloud storage applications. Hence, we adopted it to construct our systems. Over time, more powerful ABE schemes have been developed. Li [7] proposed an attribute-revocable scheme, and Chi et al. [5] proposed a policy-hiding scheme to protect data owners' privacy further. In addition, most ABE schemes involve bilinear pairing operations, which are very time-expensive, especially for resource-restricted devices such as mobiles and IoT devices. Han et al. [6] proposed a decentralized scheme to reduce the burden of data users by outsourcing the corresponding computational tasks.

### 2.2. Searchable Encryption

The main characteristic of searchable encryption (SE) is it allows users to search over many encrypted data without the decrypting of the documents in the dataset. High-level concepts of SE are that data owners extract keywords from plaintext files to build a "Secure Index" and then encrypt plaintexts with symmetric encryption schemes. Data owners transform searching keywords into corresponding trapdoors afterward. Finally, cloud servers match the Secure Indexes with the trapdoors to produce search results containing the target keywords the user longs for.

For this purpose, there are many ways to build the pre-described Secure Index. Most of the existing SE schemes involve calculating the term frequency-index document frequency (TF-IDF) values of keywords. Cao et al. [28] and Tzouramanis et al. [12] both use the K-nearest neighbors (KNN) method to build the Secure Index. It is effective; however, the associated neighborhood-related matrix will be too large, and therefore, the associated operations become time-consuming when too many keywords are involved in the system. Other methods include secure random masking, tree-based, and secure linked-list ones. The scheme proposed by Zhang et al. [25] used the secure linked-list method to build an index table, which we also adopted in our work for its efficiency.

Many functional search schemes have been developed to provide a more powerful search capability. For example, Wang et al. [13] proposed a tree-based method to provide range search. It is especially suitable for numerical datasets such as financial records. Aritomo et al. [29] and Fu et al. [10] both achieved semantic-based searching, while Zhang et al. [15] provided an efficient predicate search. Liu et al. [11] proposed a robust scheme combining semantic and fuzzy searches using fingerprint methods, which will also be adopted in our schemes. However, this scheme did not take any access control mechanism into account. They used fully homomorphic encryption (FHE) schemes [30–32]

to encrypt the index table instead. Due to complexity considerations, our work has not considered FHE schemes in our current system implementation. However, FHE schemes have lots of potential for constructing effective ABE schemes if the required complexity can be handled properly. An FHE-based ABE approach is exciting and can reduce the storage requirement of ciphertexts. We choose to put it into our future investigations.

### 2.3. Searchable ABE Schemes

Many ABE schemes have searching abilities. For this kind of scheme, it is crucial to allow only the qualified files to be searched. Otherwise, malicious users may launch keyword attacks to guess the contents of files and breach privacy. On the other hand, it is a waste of time for users to decrypt those unqualified files with failure. Sun et al. [22] proposed a famous searchable attribute scheme (ABKS) to hide the access policy. However, they use AND GATE as the access structure for policy hiding, which limits the access policy's expressiveness. Wang et al. [23] proposed a scheme that is aimed at E-health applications. They achieve a constant computational overhead, constant storage overhead, and policy hiding by hashing user attributes and keywords. However, the access policy's flexibility and searching are restricted due to its data structures. Moreover, they directly embed keyword hashes into ciphertexts, so it takes much time to match search results when there are many files in the dataset or only a single keyword can be used at a time. Miao et al. [21] and Sun et al. [33] proposed ABKS schemes with the ability for attribute revocations. Nevertheless, the searching capabilities of these schemes are weak because users can only use a single keyword once without any modifications to protocols.

## 3. Preliminaries

### 3.1. Bilinear Pairing

Following the definitions in [33], let $\mathbf{G}$ and $\mathbf{G_T}$ be two multiplicative cyclic finite groups of prime order $p$. Let $\mathbf{g}$ be a generator in $\mathbf{G}$. The following equations hold to fulfill the definition of the bilinear pairing equations.

1. Bilinearity: For all $x$, $y \in \mathbf{G}$ and all $s, t \in \mathbf{Z_p}, e(x^s, y^t) = e(x, y)^{st}$ holds. That is, the exponentiation operations inside pairings can be moved outside directly.
2. Non-degeneracy: $e(g, g) \neq 1$.
3. Computability: For all $x, y \in \mathbf{G}, e(x, y)$ and any additive or multiplicative operations on it can be efficiently computed.

### 3.2. Access Structure

By definition in [4]: Let $\{P_1, P_2, \ldots, P_n\}$ be a set of parties. A collection $\mathbf{A} \subseteq 2^{\{P_1, P_2, \ldots, P_n\}}$ is monotone if $\forall$ $\mathbf{B}$, $\mathbf{C}$: if $\mathbf{B} \in \mathbf{A}$ and $\mathbf{B} \subseteq \mathbf{C}$ then $\mathbf{C} \in \mathbf{A}$. An access structure (respectively, monotone access structure) is a collection (respectively, monotone collection), A, of non-empty subsets of $\{P_1, P_2, \ldots, P_n\}$, i.e., $\mathbf{A} \subseteq 2^{\{P_1, P_2, \ldots, P_n\}}/\varnothing$. The sets in $\mathbf{A}$ are called the authorized sets, and the sets not in $\mathbf{A}$ are called the unauthorized sets.

### 3.3. Linear Secret-Sharing Schemes

We choose the linear secret-sharing schemes as our access structure due to their full expressiveness in the access policy. Some papers [16,18,19,22,23] use the AND gate to bring efficiencies and policy-hiding capabilities. However, they do not apply to disjunctive operators. Thus, the flexibility of the access policy is quite limited.

The definition of a linear secret-sharing scheme can be found in [34]:

**Definition 1.** *Linear Secret-Sharing Schemes (LSSS)*

*A secret-sharing scheme $\mathbf{\Pi}$ over a set of parties $\boldsymbol{P}$ is called linear over $(\mathbf{Z_p})$ if*

1. *The shares for each party form a vector over $\mathbf{Z_p}$.*
2. *There exists a matrix $\boldsymbol{M}$ with is the vector of rows and n columns called the share-generating matrix for $\mathbf{\Pi}$. For the i-th row of M, we let the function $\rho$ define the party labeling row i, for all $i = 1, \ldots, l$, as $\rho(i)$. When we consider the column vector $v = (s, r_2, \ldots, r_n)$, where*

$s \in \mathbf{Z}_p$ *is the secret to be shared, and* $r_2, \ldots, r_n \in \mathbf{Z}_p$ *are randomly chosen, then* $M \cdot v$ *is the vector representing the l shares of the secret s according to the scheme* $\mathbf{\Pi}$. *The share* $(\mathbf{M} \cdot \mathbf{v})_i$ *belongs to party* $\rho(i)$.

### 3.4. Relevance Score

We use the TFxIDF measurement to express the relevance between the keyword, $w$, and the document, $F$, which has been widely adopted in many data mining and searchable encryption schemes. Term frequency (TF) represents the frequency of a keyword in the file. Nevertheless, only TF values are insufficient because some common words, such as prepositions, usually differ from what users want to search for, even if they have high occurrence frequencies in the text. Index document frequency (IDF) brings the solution. Engaged readers can find the definitions of TF and IDF in [11].

## 4. Problem Definitions

### 4.1. Threat Model

There are several players (or parties) in the investigated systems. Their role and the threat model are listed below.

**Central Authority (CA):** The central authority (CA) sets up the system and verifies intermediate user keys obtained from attribute authorities. After that, the CA produces the final user keys based on the master key generated by itself. In addition, the CA delivers the public key to the other parties. Notice that the CA is believed to be entirely trustworthy in most schemes and our systems.

**Attribute Authority (AA):** An attribute authority (AA) is equipped with some necessary cryptographic techniques, accepting the request of data users to generate user keys. They verify and generate intermediate user keys according to the attributes the data users provided. Their behavior is also honest so that they do not misbehave in the process of KeyGen and will not collide with data users.

**Data Owner (DO):** Data owners may be patients in a medical application. They extract some keywords from their medical records to build the Secure Index. After that, they upload encrypted data and the Secure Index to the cloud server. We also assume that DOs are fully credible. They will correctly extract keywords and perform succeeding encryption to the accessible files themselves.

**Cloud Server (CSP):** The cloud server provides storage to the encrypted files and performs encryption-domain searches. Their threat model is assumed to be honest but curious once again. They will honestly execute protocols but may attempt to obtain documents and keywords in plaintext form through statistical analyses. They are also interested in finding trapdoors uploaded by users, trying to guess what users are searching for, and tracing their search records.

**Data User (DU):** Data users may be doctors or researchers in an E-health application scenario. They request the encrypted files by transforming the searching keywords into respective trapdoors to perform searching. They may want to access or guess the contents of unqualified data by selective keyword attacks. However, they do not leak decrypted data to other unauthorized users.

### 4.2. System Architecture

Figure 3 shows the players, the functional blocks, and the detailed information flow of the proposed system. From Figure 3, nine polynomial-time algorithms (PTAs), as listed below, compose our system. Table 1 demonstrates the symbols used in this write-up.

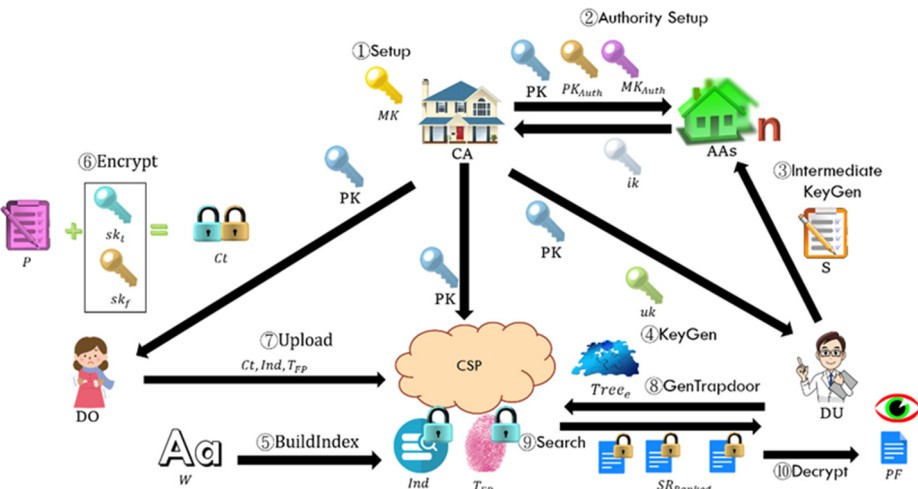

**Figure 3.** The players, the functional blocks, and the detailed information flow of the proposed system.

**Table 1.** The symbols and their corresponding definitions.

| Symbols | Description | Symbols | Description |
| --- | --- | --- | --- |
| MK | Master secret key | w | Searching keyword |
| PK | Public key | W | Keyword set |
| $MK_{Auth}$ | Authority master key | PF | Plaintext files |
| $PK_{Auth}$ | Authority public key | F | A document |
| $uk$ | User secret key | FP | Fingerprint |
| $ik$ | Intermediate user secret key | CT | Ciphertexts |
| $sk_f$ | Session key | $R_{Score}$ | Relevance score |
| $U$ | The universe of user attributes | $Str_{Search}$ | Search condition string |
| S | User attribute set | $T_{FP}$ | Fingerprint table |
| $x$ | An attribute | Td | Trapdoor |
| $H(.)$ | Hash function | $Tree_p$ | Search tree (plaintext) |
| P | Access policy | $Tree_e$ | Search tree (encrypted) |
| uid | User id | $k$ | Maximum size of searching results |
| aaid | Attribute authority id | SR | Searching results |
| Ind | Secure index | $SR_{Ranked}$ | Ranked searching results |

**Setup** $(1K, U) \rightarrow (PK, MK)$: The **CA** runs the setup algorithm and generates the master key pair. It delivers the public key, PK, to the other parties and keeps the master key, MK, for itself.

**Authority Setup** $(aaid, MK) \rightarrow (MK_{Auth}, PK_{Auth})$: The **CA** executes the authority setup algorithm to set up all the **AA**s. It grants authority to the master key, $MK_{Auth}$, and authority to the public key, $PK_{Auth}$, for each **AA**.

**IntermediateKeyGen** $(PK, uid, S, MK_{Auth}, PK_{Auth}) \rightarrow ik$: The **AA** verifies the user attribute set, S, and runs the intermediate key generation algorithm to generate the intermediate user secret key, $ik$, using its authority keypair.

**KeyGen** $(PK, MK, S, ik) \rightarrow uk$: The **CA** verifies the validity of the intermediate user key, $ik$, and then generates the final user secret key, $uk$, by the key generation algorithm.

**BuildIndex** $(PK, W) \rightarrow (Ind, TFP)$: **DO**s build an index table for each keyword, $w$, in the keyword set, W. In addition, they run a fingerprint generation algorithm to support fuzzy matching and build a fingerprint lookup table as one of the outputs. Figure 4 shows the data structure used to construct our index table.

**Encrypt** $(PK, P, W, sk_f, sk_t) \rightarrow Ct$: **DO**s extract keywords from the plaintext to obtain the keyword list, W, and then input the public key, PK, access policy, P, and the session key, $sk_f$, to the encrypted algorithm for generating the ciphertext. Finally, it encrypts the

tables with sk$_t$. **DU**s recover the session keys and decrypt files and tables associated with this ciphertext.

**GenTrapdoor** (PK, Str$_{Search}$, $uk$) $\rightarrow$ Td: **DU**s use the user key, $uk$, the public key, PK, and the search condition, Str$_{Search}$, to generate the trapdoor, Td, based on the trapdoor-generating algorithm. This algorithm has two phases: **DU**s obtain the hash values of the most proper keywords using the fingerprint-matching algorithm in the first phase. A search tree, Tree$_p$, is constructed according to Str$_{Search}$ and the hash values. Each keyword, W$'$, in Tree$_p$ is converted into a corresponding trapdoor, Td. In the second phase, all leaf nodes in Tree$_p$ are replaced by Td to become an encrypted search tree, Tree$_e$.

**Search** (Tree$_e$, Ind, $k$) $\rightarrow$ SR$_{ranked}$: The **CSP** parses the encrypted search tree, Tree$_e$, and executes the search algorithm to match *Td* with *Ind* to obtain the searching result, SR. The **CSP** sorts SR and outputs the top-k files as the final search result, SR$_{ranked}$. In our enhanced scheme, the **CSP** only matches the trapdoor, leaving the jobs of traversing searching trees and ranking for **DU**s to ensure better data privacy.

**Decrypt** ($uk$, Ct, SR$_{ranked}$) $\rightarrow$ PF: **DU**s input their user key, $uk$, ciphertext, Ct, and the ranked searching result, SR$_{ranked}$, to the decryption algorithm to obtain the plaintext files, PFs.

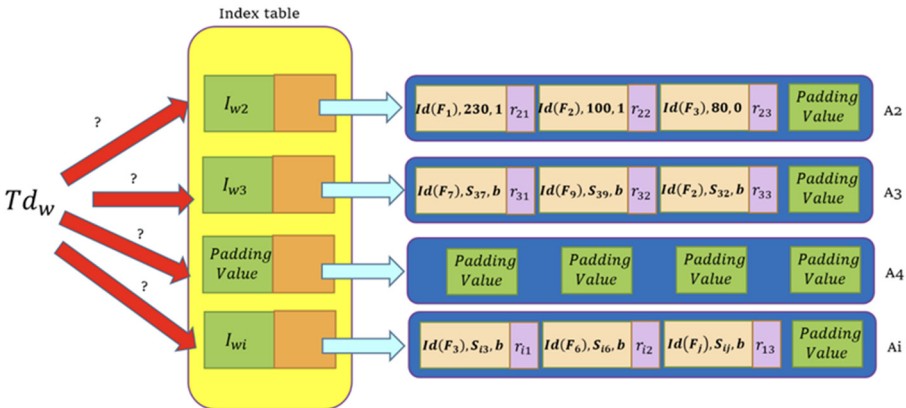

**Figure 4.** We use a link-list structure to construct our Index Table.

### 4.3. Security Model

The security model of the proposed system is built on general bilinear map cryptographic techniques and the associated assumptions. As addressed in the following paragraphs, we designed a security game to explore our system's security level. It shows that our system can defend against selective ciphertext-policy and chosen-plaintext attack (**IND-SCP-CPA**).

**The Ciphertext-domain Keyword Privacy Game**.

**Init**: Firstly, A delivers the challenge access matrix $A^*$ to $B$.

**Setup**: $B$ runs the same setup algorithm in the keyword private game.

**Phase I**: $B$ provides an oracle, $O_{SK_u}$, for a query. Furthermore, $B$ builds a secret key list, $Lst_{SK}$, to hold the query results. The oracle functions as follows:

$O_{SK}(uid, S)$: A submits $uid$ and the user attribute set, $S$, to obtain the corresponding user key, $SK_{uid,S}$. Notice that $S$ sent by A cannot satisfy the access structure, $A^*$. If $SK_{uid,S}$ has been in the keyword list, $Lst_{SK}$, $B$ looks up the list and returns the result directly. Otherwise, $B$ executes the key-generating algorithm and inserts the result into the list.

**Challenge**: A prepares two equal-length messages, $m_0$ and $m_1$, for the challenge. $B$ then decides on a random bit, $b \in 0, 1$, and encrypts them under $A^*$. Finally, $B$ sends back the ciphertext, $CT^*$, to A.

**Phase II**: $B$ can continue to query for ciphertexts after receiving $CT^*$. The operation is the same as **Phase I**.

**Guess**: A makes a guess, $b'$, for if the bit, $b$, is 0 or 1. If $b = b'$, A wins the security game.

The advantage of A to win the security game is $Adv_A = \left| Pr[b' = b] - \frac{1}{2} \right|$. Our system is IND-SCP-CPA secure if all polynomial-time adversaries only have negligible advantages at most in the security game above.

## 5. Concrete Construction

*Construction of the Basic FERMSABE Scheme*

With the pre-described nine PTAs, the basic FERMSABE system can be constructed as follows.

**Step 1**. The **CA** sets up the security parameter, K, and the global parameters $(G_1, G_T, e)$, where pairing operations e: $G_1 \times G_1 \to G_T$. Then, the **CA** generates three generators, $g$, $g_0$, and $g_1$, for the finite group, $G_1$. The **Setup** algorithm randomly chooses $a_0$, $a_1$, $b_0$, and $x$ from the group $\mathbf{Z_p}$ and chooses $v_x$ for each attribute in the universe. The rest of the public and the master keys are organized as follows.

$$PKg,\ g_0,\ g_1,\ Y = e\,(g,\ g)^x,\ A = g_0^a,\ B = g_0^b,\ \left\{ H_x = g^{b_0 \cdot v_x} \right\}_{x \in U}$$

$$MK:\ a_0,\ a_1,\ b_0 x,\ \{v_x\}_{x \in U}$$

After that, the **CA** publishes the master key pair to other parties. The **CA** further defines a hash function, $H(x) : \{0,1\}^* \to \mathbf{Z_p}$, to map keywords into elements of $\mathbf{Z_p}$.

**Step 2**. The **CA** sets up each **AA** and grants the authority key pair, $PK_{Auth}$ and $MK_{Auth}$, to the authority with an identifier, *aaid*. The **AuthoritySetup** algorithm generates a random element, t, from $\mathbf{Z_p}$ while the authority key pair comprises $PK_{Auth} \triangleq g^t$ and $MK_{Auth} \triangleq t$.

**Step 3**. When a user requests the user key, the corresponding **AA** runs the **IntermediateKeyGen** algorithm to generate the intermediate user keys using his authority key pair. The **AA** randomly picks an $\alpha$ from $\mathbf{Z_p}$ and sends this value to the **CA**. The intermediate user key, *ik*, is generated as: $ik \triangleq K_0' = \left(g^t\right)^{a_0}$ and $K_1' = \left(g^t\right)^{\alpha}$. The **AA** sends this value to the **CA** to generate the final user key.

**Step 4**. The **CA** verifies the validity of the intermediate user key, *ik*, and then uses it to run the **KeyGen** algorithm for generating the final user secret key set, *uk*, which is composed of seven components. Then, the **CA** chooses $\mu_0$ and $u$ from $\mathbf{Z_p}$. The first six components of *uk* are:

$$K_0 = g^{x_1} \cdot K_0',\ K_1 = \left(K_1'\right)^{(1/\alpha) \cdot u},\ K_2 = g^{\mu_0},\ K_3 = u,\ K_4 = g^{x_2/u},\ \text{and } K_5 = g_0^{a_0} \cdot g_1^{\mu_0}$$

Notice that $x_1$ and $x_2$ are random elements taken from $Z_p$ such that $x_1 + x_2 = x$. The **CA** generates $K_x$ for each attribute in $S$, that is $K_x = H_x^{\mu_0}$. The final user key = $K_0, K_1, K_2, K_3, K_4, K_5, \{K_x\}_{x \in S}$ and will be sent back to the data user.

**Step 5**. **DO**s build an index table, **Ind**, based on keywords extracted from plaintext files. Our **BuildIndex** algorithm is founded on the approach presented in [35] to build our **Ind**. Figure 4 depicts the data structure of our index table, where each field in blocks of the linked list represents:

- $\text{Id}(F_j)$: The identifier of the file, $j$, which contains the keyword, $i$.
- $S_{ij}$: The relevance score of the keyword, $i$, and the file, $j$. Notice that the blocks will not be sorted according to this score for confusion.
- $r_{ij}$: Random strings of the same length. We use this field to prevent producing two identity blocks.
- Padding values: We add padding values to every linked list to make them of the same size. This setting implies that some linked lists composed of all padding values may also be appended to the table.

Furthermore, **DO**s build a fingerprint table to support fuzzy search. Figure 5 illustrates the structure of our fingerprint table, and the corresponding generation algorithm can be found in [15]. We store the hash value of a keyword instead of itself to prevent **DU**s from

knowing the keywords of **DO**s directly. Only the hash value is enough for the subsequent matching and searching tasks.

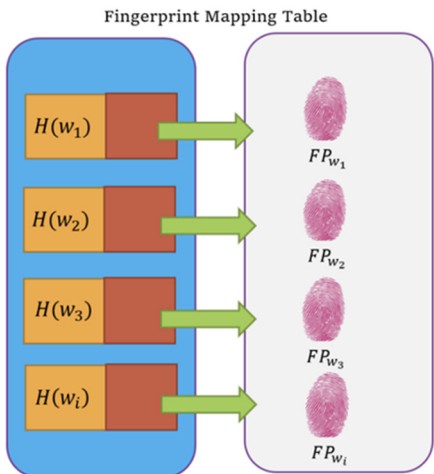

**Figure 5.** Data Structure of our Fingerprint Mapping Table.

In addition to these tables, the **DO** needs to put some extra data into the headers of **Ind** to allow the cloud server to perform matchings. We list the additional information in the following:

$$I_0 = g_1^s, \ I_2 = e(A_0, \ g_0)^s, \ \left\{ I_{1,x} = H_x^{\left( \frac{1}{H(w)} \right)} \right\}_{x \in \rho(i)}, \ \text{and } I_3 = B^{s/H(w)}. \tag{1}$$

Finally, the **DO** uploads the encrypted **Ind** and ciphertexts to the cloud server.

**Step 6**. **DO**s extract keywords from the plaintext files, **PF,** to build the keyword list, W, and input the public key, PK, access policy, P, and the session keys, $sk_f$ and $sk_t$, to the **Encrypt**

Algorithm. The former is used to encrypt **PF**, and the latter is used to encrypt **T$_{FP}$** by symmetric encryption algorithms such as AES. They choose two elements, $s$ and $s'$, from $Z_p$ for supporting secret sharing and, respectively, build the secret sharing vectors, $\lambda_x$ and $\lambda_x'$, for $x \in \rho(i)$ by LSSS schemes as follows. They further compute

$C_0 = sk_f \cdot e(g,g)^{x \cdot s}, \ C_1 = g^s, \ \left\{ C_x = g^{a_0 \cdot \lambda_x} \right\}_{x \in \rho(i)}, \ C_2 = sk_t \cdot e(g,g)^{x \cdot s'}, \ C_3 = g^{s'}, \ \text{and}$

$\left\{ D_x = g^{a_0 \cdot \lambda_x'} \right\}_{x \in \rho(i)}$. Finally, **DO**s upload $\{ \{ \text{CT} = C_0, \ C_1, \ C_2, \ C_3, \ \{C_x\}, \ \{D_x\} \}, \text{Enc}_{sk_f}(\textbf{PF}),$ $\text{Enc}_{sk_t}(\textbf{TF}_\textbf{P}), \ \text{Enc}_{sk_t}(\textbf{Ind}) \}$ to **CSP**.

**Step 7**. **DU**s first download the ciphertext pack from **CSP** and decrypt **Ind** and $T_{FP}$ with uk by the **Decrypt** algorithm. If **DU**s own the right user key, $sk_t$ can be obtained to decrypt these tables correctly. Otherwise, the algorithm halts. By using a fuzzy matching algorithm, **DU**s can find the fingerprint that best matches the fingerprint of the input keyword, where we adopt the fuzzy matching algorithm presented in [15] to realize this function. Nevertheless, we additionally set a matching threshold to 0.7. Suppose the relevance score between the best-matched fingerprint and the query fingerprint is lower than this threshold, the match will be discarded, and the corresponding leaf node will be removed to prevent fetching unrelated documents. Second, **DU**s look up **TF$_P$** to obtain the best-matching hash value, $H(w')$. After that, **DU**s parse **Str$_{\textbf{Search}}$** to build a search tree, as shown in Figure 6. Finally, **DU** chooses a random element, $\gamma_u$, from $Z_p$ to disturb all the values on the leaf nodes. That is, using the **GenTrapdoor** algorithm, we compute

$$T_0 = K_2 \cdot g^{\gamma_u} \ \text{and} \ T_1 = K_{5 \cdot g_1^{\gamma_u}} \cdot \sum_{x \in S} (K_x \cdot H_x)^{1/H(w')}$$

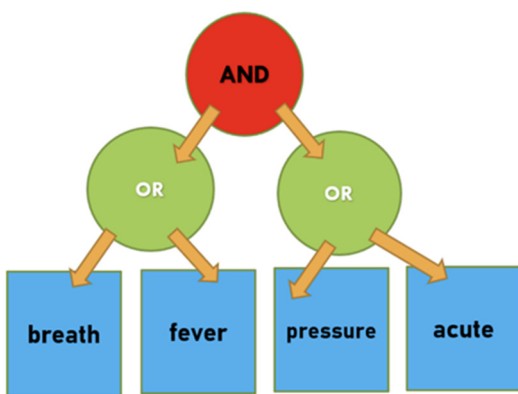

**Figure 6.** The Query keyword tree in plaintext form. This table is generated for the access condition of (**breath OR fever**) **AND** (**pressure OR acute**). Notice that this figure is for demonstration purposes only. In actuality, **DU**s need not know which keywords they have precisely matched.

**DU**s replace the plaintext domain to-be-searched keywords with these two values at the corresponding locations to produce $\mathbf{T_W}$ for searching. Eventually, **DU**s provide $\mathbf{T_W}$ and the decrypted **Ind** to the cloud server.

**Step 8**. The **CSP** first parses the encrypted search tree, $\mathbf{Tree_e}$. Then, it matches each **Td** in $\mathbf{Tree_e}$ with each header information in **Ind**. In other words, it compares whether $I_2 \cdot e(T_0, I_0 \cdot \Pi_{x \in S} I_{1,x}) = e(C_1, T_1)$? If any index satisfies the previous condition, all the document indexes stored in the latter linked list will be appended to the tree node. Notice that we only need to compute the right-hand side term once because it is fixed. Therefore, our **Search** algorithm is quite efficient. After all the leaf nodes are searched, the **CSP** takes the intersection or union of the search results' leaf nodes to become the final search results of the parent nodes depending on whether their parents are **AND** node or **OR** node. Finally, the **CSP** sorts the searching results, **SR,** in the root node and outputs the top-k files as the final searching result, $\mathbf{SR_{ranked}}$. Then, $\mathbf{SR_{ranked}}$ will be sent back to the **DU**s.

**Step 9**. In the final phase, **DU**s use their user keys, $uk$, to match with the ciphertext, Ct, for finding the decryption keys. **DU**s will compute $E = \frac{\prod_{x \in S} e(C_x, K_1)^{\omega_x}}{e(C_1, K_4)} = \frac{e(g,g)^{\frac{\alpha_{st}}{\mu}}}{e(g,g)^{\frac{sx_2}{\mu}}}$.

Using $E$, they further compute $R = \frac{C_0 \cdot E^{K_3}}{e(g^s, K_0)}$.

Suppose the user key satisfies the access policy. In that case, **R** will be identical to the final decryption key, $sk_f$. Finally, **DU**s can use this key to decrypt encrypted data retrieved in the previous step and obtain the plaintext files. We will present the correctness proofs of searching and decryption in the next Section.

## 6. Analyses

### 6.1. Security Analyses

In this section, we explore the proofs of the security model as mentioned above and other functional modules of our system.

Theorem 1: Assume the $q$-parallel bilinear Diffie–Hellman ($q$-BDHE) assumptions hold in both **G** and $\mathbf{G}_T$ groups. There is no probability that any polynomial-time adversary, A, can break the security of our schemes with a non-negligible advantage.

Proof: Assume the advantage of distinguishing a valid ciphertext from a random element for $A$ is $\varepsilon_1 = Adv^{IND-sCP-CPA}$. We built a simulator, B, that can break the $q$-BDHE assumption with a non-negligible advantage $\varepsilon_1/2$.

The $q$-BDHE challenger, C, first selects random elements $a, s, b_1, \ldots, b_q$ from $\mathbf{Z}_p$ and sets

$$\varphi = \left( g, g^s, \ldots, g^{a^q}, g^{a^{q+2}}, \ldots, g^{a^{2q}}, g^{s \cdot b_j}, g^{a/b_j}, g^{a^{q/b_j}}, g^{a^{q+2/b_j}}, \ldots, g^{a^{2q/b_j}}, g^{a \cdot s \cdot b_i/b_j}, \ldots, g^{a^q \cdot s \cdot b_i/b_j} \right).$$

According to the definition of $q$-BDHE, A is still hard to distinguish $e(g, g)^{a^{q+1} \cdot s}$ even if

he knows the above arguments. Then, C chooses a random bit, $\gamma \in 0, 1$. If $\gamma = 0$, C sets $T = e(g, g)^{a^{q+1} \cdot s}$. Otherwise, $T$ is set to a random element in $G_T$.

**Init**: The simulator, B, received a $q$-BDHE challenge instance $(\varphi, T)$. The adversary, A, announces a challenge access structure $(M^*, \rho^*)$ and sends it to B, where $M^*$ is an $l^* \times n^*$ matrix and $l^*, n^* < q$.

**Setup**: B selects an element, $x'$, in $Z_p$ randomly and sets $e(g, g)^x = e\left(g^a, g^{a^q}\right) \cdot e(g, g)^{x'}$ which implicitly makes $x = x' + a^{q+1}$. In addition, B initializes a $v_x$ for each attribute by choosing $v_x \in Z_p$ at random, and also randomly selects an element, $b_0$, from the same group. Finally, B sets $H_x = g^{b_0 \cdot v_x}$ and gives the partial public key parameters to A.

**Phase I**: B keeps a list of the tuple $(uid, S, SK)$ represented as $\text{Lst}_{SK}$. Initially, the list is empty. A can query the following oracle in the polynomial form:

- $O_{SK}(uid, S)$: Assume that B received a secret key query for $(uid, S)$, in which $S$ does not match the access structure $(M^*, \rho^*)$. B performs the following operations: if A has previously asked for $S$, B retrieves $SK$ from the list, $\text{Lst}_{SK}$, directly and returns it to A.

  Otherwise, B chooses a vector, $\gamma = (\gamma_1, \ldots, \gamma_{N^*}) \in Z_p$, such that $\gamma_1 = -1$ and $M_i^* \cdot \gamma = 0$ for all $i, \rho^*(i) \in S$. This matrix must exist according to the properties of LSSS. Then, B randomly picks $\sigma \in Z_p$ and represents t as: $t = \sigma + \gamma_1 a^q + \gamma_2 a^{q-1} + \ldots + \gamma_n a^{q+1-n^*} \cdot$B further selects $x_1'$, $x_2' \in Z_p$ at random, such that $x_1' + x_2' = x' \mod p$, and sets $x_1 = x_1' + a^{q+1}$ and $x_2 = x_2'$. Then, B, respectively, calculates $K_1$ and $K_4$ as: $K_1 = g^{\frac{\sigma}{\mu}} \cdot \prod_{i=1,\ldots,n^*} \left(g^{a^{q+1-i}}\right)^{\frac{r_i}{\mu}} = g^{\frac{t}{\mu}}$ and $K_4 = g^{x_2/u} = g^{x_2'/u}$. Through the definition of $t$, we noticed that $g^{at}$ contains a term of $g^{a^{q+1}}$, which can be ignored with the unknown terms in $g^{x_1}$ when calculating $K_0$. That is, B computes $K_0$ as: $K_0 = g^{x_1'} g^{\alpha\sigma} \cdot \prod_{i=2,\ldots,n^*} \left(g^{a^{q+2-i}}\right)^{\gamma_i} = g^{x_1} \cdot g^{at}$. Notice that $K_5$ and $K_x$ are irrelevant to $t$, $x_1'$, and $x_2'$, so we omit the generation of them here. Finally, B puts $SK = \{K_0, K_1, K_2, K_3, K_4, K_5, \{K_x\}_{x \in S}\}$ into $\text{Lst}_{SK}$ and sends the keys to A.

**Challenge**: A prepares two equal-length messages, $m_0$ and $m_1$, for the challenge. B then decides on a random bit, $b \in 0, 1$, and encrypts them under $M^* \cdot$B computes $C_0^*$ as $C_0^* = m_b \cdot T \cdot e(g^s, g^x)$, and $C_1^*$ is generated as $C_1^* = g^s$.

It is hard for B to simulate $C_x^*$ since it includes the term $g^{a^j s}$. To overcome this difficulty, B splits the secret to eliminate the above-mentioned terms. That is, B selects $y_2', \ldots, y_{n^*}' \in \boldsymbol{Z_p}$ randomly, and then shares the secret vector, $V = \left(s, sa + y_2', sa^2 + y_3' + \ldots + sa^{n^*-1} + y_{n^*}'\right) \in \boldsymbol{Z_p}$, with A. For $i \in [1, l]$, we describe $Q_i$ as the set of all $k \neq i$ making $\rho^*(i) = \rho^*(k)$. B calculates $C_{x^*}$ as: $C_{x^*} = \prod_{i=2,\ldots,n^*} (g^a)^{M_{i,k}^* \cdot y_k} \cdot \prod_{x \in Q_l, k=1,\ldots,n^*} \left(g^{\frac{a_j s b_i}{b_l}}\right)^{-M_{i,k}^*}$.

We produce $C_2^*$, $C_3^*$, and $D_x^*$ in the similar way. Finally, B returns the challenge ciphertext, $C_T' = \left\{C_0^*, C_1^*, C_2^*, C_3^*, \{C_i^*, D_i^*\}_{i \in [1,l]}\right\}$, to A.

**Phase II**: A continues to make queries similar to **Phase I**.

**Guess T**: A outputs $b'$ which is a guess of $b$. If $b' = b$, B returns $\gamma = 0$ to guess $T = e(g, g)^{a^{q+1} \cdot s}$. Otherwise, B returns $\gamma = 1$, indicating that $T$ is a random element chosen from $G_T$. In this case, A won the security game and obtained an effective ciphertext. Now, the advantage of A is $Pr[b' = b | \gamma = 0] = 1/2 + \varepsilon_1$. Conversely, A cannot obtain any information about b and the ciphertext; thus, $Adv_B = 1/2$. In conclusion, the advantage of A in winning the IND-SCP-CPA security game is: $\frac{1}{2}\left(\frac{1}{2} + \varepsilon_1\right) + \frac{1}{2} \cdot \frac{1}{2} - \frac{1}{2} = \frac{\varepsilon_1}{2}$. Since A only has a negligible advantage in solving the $q$-DBHE problem, hence no polytime adversary, A can break the security of our schemes with a non-negligible advantage.

As for the keyword privacy, we will prove that any polytime adversary, A, cannot guess the input keyword, w, from the Secure Index, I, nor forge it.

Firstly, because the secret value, s, masked the term $I_3 = g^{\frac{(b_0 \cdot s)}{H(w)}}$. Even if A has produced the value $g^{1/H(w)}$ on its own, the only term which contains $b_0$ is $I_{1,x} = g^{\frac{(b_0 \cdot v_x)}{H(w)}}$. A cannot

obtain the value, $v_x$, because it is one of the components of the master key, *MK*, to tell or forge the Secure Indices. To change the keyword of a trapdoor, A needs to modify $T_1 = K_5 \cdot g_1^{\gamma_u} \cdot \prod_{x \in S} \left( K_x \cdot H_x^{\gamma_u} \right)^{1/H(\omega')}$. However, it is hard due to the difficulty in solving the discrete log problem.

In summary, the unmalleability of the index and trapdoor of our scheme has now been proved.

### 6.2. Functional Comparisons

We compared some existing ABKS schemes with ours in terms of access control, keyword search, multi-keyword, ranked result, fuzzy search, and semantic search capabilities, as shown in Table 2. We use the symbol "✓ " to mean that the scheme has the indicated function, while the symbol "-"represents the lack of this kind of function. Our scheme is the most functional from the table, providing fine-grained access control and supporting a multi-keyword ranked search result with various powerful search modes.

**Table 2.** Functional Comparison between the proposed and the benchmarked ABKS schemes.

| Function | MABKS [21] | MSDVABE [33] | FSSE [11] | Ours |
|---|---|---|---|---|
| Access control | ✓ | ✓ | - | ✓ |
| Keyword search | ✓ | ✓ | ✓ | ✓ |
| Multi-keyword | - | ✓ | ✓ | ✓ |
| Ranked result | - | - | ✓ | ✓ |
| Fuzzy search | - | - | ✓ | ✓ |
| Semantic search | - | - | ✓ | ✓ |

### 6.3. Computational Complexity Analyses

Table 3 compares the theoretical computation costs with some recent **ABKS** schemes and ours. Let |U| denote the universe size and |S| the size of user attributes, while we use |L| to represent the number of attributes the **DO** used in the access policy. We use P to symbolize pairing operations. E and $E_t$ represent the exponentiation operations in groups $G$ and $G_T$. Hash functions are excluded from our comparison because they are much more efficient than exponentiation and pairing operations. The table shows that our scheme is the most efficient one most of the time, especially for searching.

**Table 3.** Comparisons of Theoretical Computational Costs Between Our Scheme and the Benchmarked Ones.

| Function | MABKS [21] | MSDVABE [33] | OABRSE [11] | Ours |
|---|---|---|---|---|
| Setup | $(\|U\|+3)E + P + E_T$ | $(\|U\|+4)E + P$ | $(\|U\|+2)E + P + E_T$ | $(\|U\|+2)E + P + E_T$ |
| KeyGen | $(3\|S\|+8)E$ | $(\|S\|+4)E$ | $(\|S\|+4)E$ | $(\|E\|+5)E$ |
| Enc | $(4\|L\|+3)E + P + 2E_T$ | $(3\|L\|+1)E + E_r$ | $(\|L\|+1)E + 1E_T$ | $(2\|L\|+2)E + 2E_T$ |
| Trap | $(2\|S\|+2)E$ | $3E$ | $4E$ | $(2\|S\|+2)E$ |
| Search | $2P$ | $3P$ | $3P$ | $2P$ |
| Dec | $(2\|S\|+1)P + \|S\|E_T$ | $(2\|S\|+1)P + 2\|S\|E_T$ | $(\|S\|+2)P + (\|S\|+1)E_T$ | $(\|S\|+2)P + (\|S\|+1)E_T$ |

We concluded our theoretical storage costs compared with the above-mentioned schemes in Table 4. $|G|$, $|G_T|$, and $|Z_p|$ are bit lengths required to store an element in the respective finite group. Our theoretical storage costs are similar to the MABKS [17] scheme. However, our scheme has lower constant terms and has little relevance to the user attribute size. Furthermore, our trapdoor size is quite reasonable compared with the other schemes. We put extra data into ciphertexts to eliminate the need for **DO**s to exchange keys with **DU**s. Even so, the space complexity of ciphertexts is still acceptable in actual cases.

**Table 4.** Comparisons of Theoretical Storage Costs Between Our Scheme and the Benchmarked Ones.

| Function | MABKS [21] | MSDVABE [33] | OABRSE [11] | Ours |
|---|---|---|---|---|
| Master keypair | $(\|U\|+6)\|G\|+\|G_T\|+$ $(\|U\|+3)\|Z_p\|$ | $(\|U\|+5)\|G\|+\|G_T\|+$ $(\|U\|+1)\|Z_p\|$ | $(\|U\|+3)\|G\|+\|G_T\|+$ $(\|U\|+2)\|Z_p\|$ | $(\|U\|+5)\|G\|+\|G_T\|+$ $(\|U\|+3)\|Z_p\|$ |
| User key | $(3\|S\|+4)\|G\|$ | $(\|S\|+4)\|G\|+\|Z_p\|$ | $(\|S\|+4)\|G\|+\|Z_p\|$ | $(\|S\|+5)\|G\|+\|Z_p\|$ |
| Ciphertext | $(2\|L\|+1)\|G\|+2\|G_T\|$ | $(2\|L\|+2)\|G\|$ | $(2\|L\|+1)\|G\|+\|G_T\|$ | $(2\|L\|+2)\|G\|+2\|G_T\|$ |
| Index | $(\|S\|+2)\|G\|+\|G_T\|$ | $3\|G\|$ | $3\|G\|$ | $(\|S\|+2)\|G\|+\|G_T\|$ |
| Trapdoor | $2\|G\|+\|Z_p\|$ | $3\|G\|$ | $3\|G\|$ | $2\|G\|$ |

*6.4. Experimental Analyses*

We designed a series of experiments to simulate the actual performance of our schemes. We used the real Enron email dataset [35] for testing. Moreover, we tested our schemes on a Windows machine with 2.80 GHz Intel(R) Core(TM) i7-1165G7 @ 2.80 GHz CPU and 8 GB ROM. We used JPBC (Java Pairing-Based Cryptography) as the pairing operation library and executed the programs on Java SDK 17 and JPBC 2.0.0. According to the most popular setting, we set $\|Z_p\| = 160$ bit and $\|G\| = \|G_T\| = 1024$ bit, and the Type-A elliptic curve: $y^2 = x^3 + x$ is picked. For practical uses, the universe size is between [20, 100], and the user attribute size is between [3, 100]. In the subsequent experiments, we assumed at least one authorized document for **DU**s to retrieve.

Figure 7a–d shows the simulation results of our basic scheme compared with others. The universe and the user attribute sizes have been mentioned above. Because some of these schemes do not support multi-keyword ranked search, we only examined one document and one searching condition for ease of simulations. However, it is sufficient to express the effectiveness of the proposed scheme. Figure 7a shows the setup time, demonstrating a linear dependency on the size of the system attributes. While the encryption time is irrelevant to the size of the system attributes, as shown in Figure 7b, our setup time is similar to the other benchmarking schemes, but we use a much shorter time for encryption. Our scheme shows superiority in decryption and user-key generation time, as demonstrated in Figure 7c,d. Notice that the required key-generation time is proportional to the size of the user attributes rather than that of the universe. Clearly, our scheme has more advantages when massive user attributes are required. Our approaches are the most efficient compared to the MABKS [21] and MSDVABE [33] schemes.

We constructed a practical system for the implementation of our enhanced scheme. Figure 8a–e shows this system's actual data retrieval and index-table building times. For ease of simulations, we realized the same extensions on the other benchmarked schemes to support more powerful searching modes. In these experiments, we set the universe size to 27, and the user attributes size to 3 for simulating real scenarios. These attributes are categorized into position, subject, and level classes. This setting does not affect the experiment results in any case. In Figure 8a, we fixed the size of the keywords.

We set the number of Provided by **DO**s to 30 and the number of search conditions selected by DUs to 5. Furthermore, we set the size of the document database to vary from 20 to 100. In this circumstance, our search time is almost constant and is similar to that of **MABKS** [21]; both are better than the **MSDVABE** [33]. In Figure 8b, the keyword size varies from 20 to 100 while the database size and searching conditions are fixed to 100 and 5, respectively. Our searching time is linearly proportional to the size of the keywords, while that of the **MSDVABE** [33] scheme varies more dramatically than ours. The same conclusion can be drawn from Figure 8. When the search conditions increase from 5 to 30, our scheme performs better than the others.

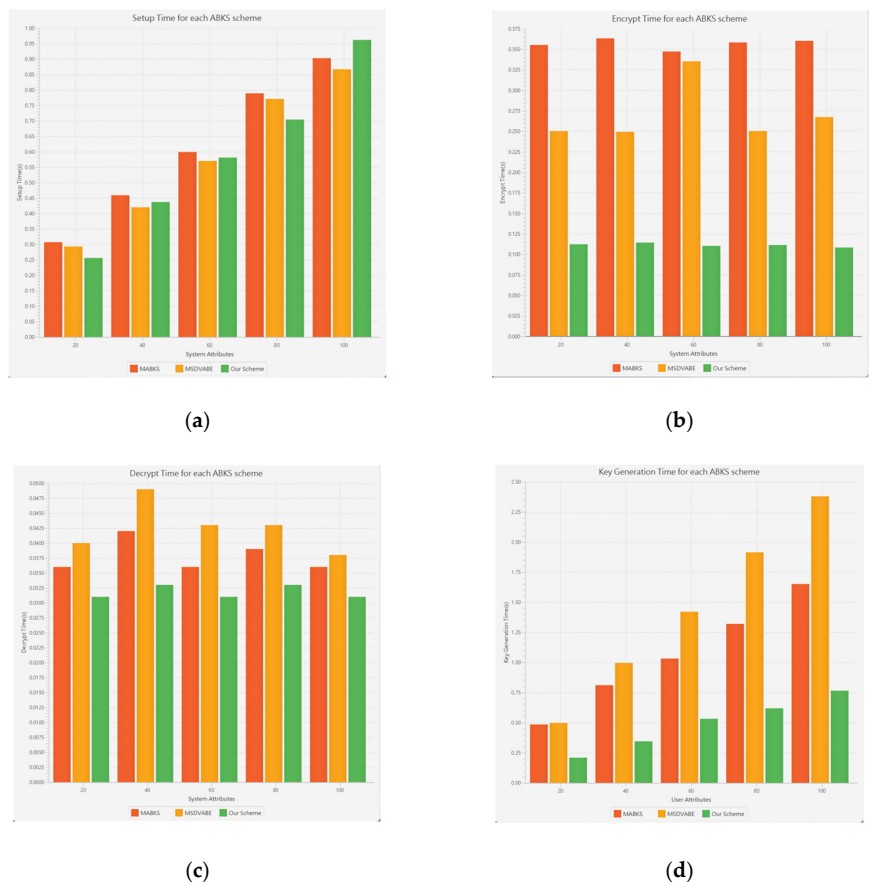

**Figure 7.** Timing Performance Comparisons. (**a**) Setup time, (**b**) Encryption time, (**c**) Decryption time, and (**d**) Time cost of user secret-key generation.

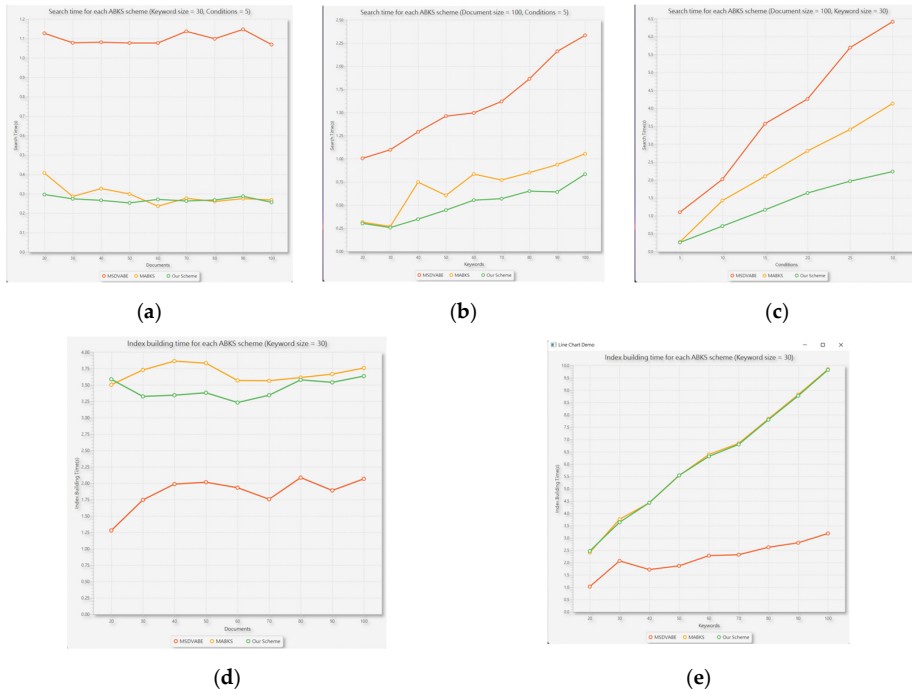

**Figure 8.** Experiment results in Our Realized Practical Systems. Data retrieval time for (**a**) different document sizes, (**b**) different keyword sizes, and (**c**) different searching conditions. Index-Table building time for (**d**) different documents and (**e**) different keyword sizes.

Figure 8d–f demonstrates the actual consuming times for building an index table. Although the **MSDVABE** [33] scheme takes the shortest time in this experiment, it has a poor performance on searching. With a similar opinion to **MABKS** [21], we conducted one pairing operation in the index-building phase to prevent performing too many pairing operations in the searching phase. Therefore, some of the performance on building index tables is sacrificed. However, data owners usually build index tables only once, but data users may search the database many times. Therefore, our schemes are most realistic and practical in actual use. Furthermore, these two schemes take much more time, even making it impossible to perform fuzzy and semantic keyword-ranked searches combined with multiple keywords without our extensions. We proved that our schemes are efficient, flexible, and universal to apply to other performance-oriented AMKS schemes.

## 7. Conclusions

In this paper, we showed that the proposed **FEMRSABE** scheme has a powerful search capability that can satisfy most users' needs. Even if the user inputs do not fully match the keywords set up by the **DO** or have some minor spelling errors, users can still obtain the desired and most-related documents. Our basic protocol competes with the state-of-the-art schemes through the performance analyses given in the previous Section.

The state-of-the-art takes much more time to search and does not perform fuzzy and semantic keyword ranked searches which is the main contribution of our work.

Moreover, the enhanced one brings many more functionalities with a slight efficiency loss, which is tolerable in real-world scenarios. Moreover, we proved that our scheme is secure under the **IND-SCP-CPA** and the **IND-CKA** security requirements. However, there are some limitations in our system as well. For example, the attributes of users may frequently vary in the real world, while fine-grained attribute revocation and updating mechanisms are needed but are not included in our work currently. Furthermore, we tackle the single-point failure problem by setting up multiple attribute authorities, but there are probably malicious attribute authorities that can determine users' privacy by mis-operations.

We plan to add the attribute revocation and verification mechanisms mentioned above to make the system more steady and secure.

**Author Contributions:** Formal analysis, J.-K.L.; Funding acquisition, J.-L.W.; Investigation, J.-K.L., W.-T.L. and J.-L.W.; Methodology, J.-K.L.; Project administration, W.-T.L. and J.-L.W.; Resources, J.-L.W.; Software, J.-K.L.; Supervision, W.-T.L. and J.-L.W.; Writing—original draft, J.-K.L.; Writing—review & editing, W.-T.L. and J.-L.W. All authors have read and agreed to the published version of the manuscript.

**Funding:** The Minister of Science and Technology, Taiwan: MOST 111-2221-E-002-134-MY3 and Taiwan Semiconductor Manufacturing Company: TSMC: 112H1002-D.

**Data Availability Statement:** Not applicable.

**Conflicts of Interest:** The authors declare no conflict of interest.

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
