# Peer review of "Flexible and Efficient Multi-Keyword Ranked Searchable Attribute-Based Encryption Schemes"

_cryptography, doi:10.3390/cryptography7020028_

Round 1

Reviewer 1 Report

Review of paper 2295294

Researching “Flexible and Efficient Multi-Keyword Ranked Searchable Attribute-based Encryption Schemes” is a very important topic in the cloud storage and IoT services because it gives to those interested in cloud computing, an idea regarding the things which must be considered when using not fully trustable cloud service providers and servers as well as Public Key Encryption. The evolution of cloud servers and access controllability are somehow connected, both manifesting challenges due to the interactions between users and the system with the management of secret keys. Encrypting the data and uploading it on servers may lack proper access controllability.

Introduction

Does the introduction provide sufficient background information for readers not in the immediate field to understand the problem / hypotheses?

Yes. The first section of the paper presents the hierarchy of standard searching modes. Also, it presents the methods used for high-level searching: beside single and multi-keyword rank search, there are fuzzy and semantic search.

Are the reasons for performing the study clearly defined?

The reasons for developing the research and its objectives for the new design of a flexible multi-keyword ranked searchable attribute-based scheme using search trees must be further defined.

Are the study objectives clearly defined?

Yes. The main objective is to demonstrate the efficiency and security of the flexible multi-keyword ranked searchable attribute-based schemes.

What is the main question addressed by the research?

Can be made a flexible multi-keyword ranked searchable attribute-based scheme to combine fuzzy searching keywords with AND-OR logic gates?

2. Literature Review and Model Development

Is the literature cited balanced or are there important studies not cited, or other studies disproportionately cited?

The cited literature in this work includes 36 titles and is related to the topics of similarity search scheme, Searchable encryption cloud storage with dynamic data, Homomorphic Encryption Scheme, privacy and security in decentralized ciphertext-policy attribute-based encryption and User collision avoidance CP-ABE with efficient attribute revocation for cloud storage.

Please identify statements that are missing any citations, or that have an insufficient number of citations, given the strength of the claim made.

-

Do you consider the topic original or relevant in the field? Does it address a specific gap in the field?

Yes. It is quite unique. The research makes remarkable progress with the proposed index-table method and the tree-based searching algorithm, we proved the efficiency and security of the proposed schemes through a series of analyses and experiments.

3. Methodology and Data

Are the methodology and data used appropriate to the purpose of the research?

Yes, it seems like it, but they may be further detailed and defined.

Is sufficient information provided for a capable researcher to reproduce the experiments described?

May be further developed.

Are any additional experiments required to validate the results of those that were performed?

Yes. Further experimentation would provide more data regarding the efficiency and security of the attribute-based encryption.

Are there any additional experiments that would greatly improve the quality of this paper?

Yes. Experimentation and a series of analyses can help.

Are appropriate references cited where previously established methods are used?

Yes

4. Results

Are the results clearly explained and presented in an appropriate format?

Yes

Do the figures and tables show essential data or are there any that could easily be summarized in the text?

Yes.

Are any of the data duplicated in the graphics and/or text?

No.

Are the figures and tables easy to interpret?

May be further improved.

Are there any additional graphics that would add clarity to the text?

Surely. The link-list structure, functional blocks, and the detailed information flow are improving the research definition.

Have appropriate statistical methods been used to determine the significance of the results?

Yes.

What does it add to the subject area compared with other published material?

The research applies the semantic search for a boost in flexibility and searching experience.

5. Conclusions and Implications

Are all possible interpretations of the data considered or are there alternative hypotheses that are consistent with the available data?

The experiments, analyses and their interpretations supported by the simulation results are offered in chapter 6. The interpretations may be further developed.

Are the findings properly described in the context of the published literature?

Yes, in the section dedicated to the Experimental Analyses, there are some indications that experiments were carried out to simulate the actual performance flexible multi-keyword ranked searchable attribute-based scheme, but further interpretations are encouraged to be made.

Are the limitations of the study discussed? If not, what are the major limitations that should be discussed?

The limitations of the study should be further described, as well as the insight in the future developments of the research.

What specific improvements should the authors consider regarding the methodology? What further controls should be considered?

The authors could use alternative tools for encrypting the data.

Are the conclusions consistent with the evidence and arguments presented and do they address the main question posed?

Yes. The conclusion may be further defined with experimental statistical data.

Are the conclusions of the study supported by appropriate evidence or are the claims exaggerated?

Conclusions are supported by the proposed scheme in the paper, demonstrating the search capability that can satisfy most users’ needs-so they say. The improvement brings much more functionalities with slight efficiency loss. The findings should be quantified

Are the references appropriate?

Yes

Significance and Novelty

Are the claims in the paper sufficiently novel to warrant publication?

Yes.

Does the study represent a conceptual advance over previously published work?

Yes, it does, by presenting the scheme for a flexible multi-keyword ranked semantic search.

Journal Selection

Is the target journal (if known) appropriate? If not, why not?

Yes

What is the likely target audience of this paper?

This scientific paper is useful mainly to cloud safety designers and server security staff who need specific data regarding encryption algorithms; to Information Engineering experts and Computer Science technicians, and Networking and Multimedia experts.

Minor comments

Please refer to the comments in the edited manuscript file for minor comments.

Accept if minor revisions are made.

Major comments

To publish this paper in your target journal, the following revisions are strongly advised:

The English language and style are fine/minor spell check required.

Additional comments on the tables and figures.

There are no additional comments.

Author Response

Thanks for reviewing and sharing valuable suggestions. Below is our point-to-point reply to the comment:

Introduction:
- Are the reasons for performing the study clearly defined?
> The reasons for developing the research and its objectives for the new design of a flexible multi-keyword ranked searchable attribute-based scheme using search trees must be further defined.

We have added an extra paragraph in the Introduction to describe our potential use cases.

Methodology and Data:
- Are the methodology and data used appropriate to the purpose of the research?
> Yes, it seems like it, but they may be further detailed and defined.

The methodology is defined in “Chapter 3 algorithm” and “Chapter 4.2 system architecture”, respectively.

- Is sufficient information provided for a capable researcher to reproduce the experiments described?
> May be further developed.

The reader can refer to Chapter 3 and Chapter 4.2 for the algorithm and system architecture in detail from our revision.

- Are any additional experiments required to validate the results of those that were performed?
> Yes. Further experimentation would provide more data regarding the efficiency and security of the attribute-based encryption.

We have included the theoretic time complexity analysis in Table 2. The practical execution time, including setup, encryption, decryption, key generation, and retrieval, can be found in Figure 7 and Figure 8. The security model is IND-sCP-CPA; the proof can be found in Chapter 6.1.

Results:
- Are the figures and tables easy to interpret?
> May be further improved.

As suggested, we have revised the descriptions for all the Figures and Tables.

Conclusion and Implications:
- Are all possible interpretations of the data considered or are there alternative hypotheses that are consistent with the available data?
> The experiments, analyses and their interpretations supported by the simulation results are offered in chapter 6. The interpretations may be further developed.

We have added more interpretations of our experiment results in Chapter 6.

- Are the findings properly described in the context of the published literature?
> Yes, in the section dedicated to the Experimental Analyses, there are some indications that experiments were carried out to simulate the actual performance flexible multi-keyword ranked searchable attribute-based scheme, but further interpretations are encouraged to be made.

As suggested, we have added more interpretations of our experiment results in Chapter 6.

- Are the limitations of the study discussed? If not, what are the major limitations that should be discussed?
> The limitations of the study should be further described, as well as the insight in the future developments of the research.

As suggested, the limitations and future research directions are added to Chapter 7.

- Are the conclusions consistent with the evidence and arguments presented and do they address the main question posed?
> Yes. The conclusion may be further defined with experimental statistical data.

As suggested, we added more experimental results descriptions in our conclusion section.

- Are the conclusions of the study supported by appropriate evidence or are the claims exaggerated?
> Conclusions are supported by the proposed scheme in the paper, demonstrating the search capability that can satisfy most users’ needs-so they say. The improvement brings much more functionalities with slight efficiency loss. The findings should be quantified.

As suggested, we added more experimental results descriptions in our conclusion section.

Major comments:
- To publish this paper in your target journal, the following revisions are strongly advised:
> The English language and style are fine/minor spell check required.

We have revised the paper's spelling as much as possible with the aid of Grammarly.

Reviewer 2 Report

What represent figures 1 and 2? The authors should add more details about the relevance of figures 1 and 2.

The original contribution is unclear. The contribution is a system or scheme FEMRSABE?

All acronyms used should be explained.

Author Response

Thanks for reviewing and sharing valuable suggestions. Below is our point-to-point reply to the comment:

What represent figures 1 and 2? The authors should add more details about the relevance of figures 1 and 2.

Figure 1 shows the summary of a standard searching model from low level to high level. For example, the basic model uses only single or multiple keywords to do the exact search. A more high-level approach, like fuzzy and semantic search, allows users to have some typos or feed synonymous. An improved description for Figure 1 is added to Section 1.1.

Figure 2 depicts the data structure of the search tree. Details are already available in lines 72-79.

The original contribution is unclear. The contribution is a system or scheme FEMRSABE?

Our main contribution is to construct multi-keyword ranked searchable attribute-based encryption schemes (FEMRSABE) with two advanced search mechanisms: fuzzy and semantic search. Furthermore, our scheme targets cloud-user scenarios, so we also provide efficient and flexible system architectures for them.

All acronyms used should be explained.

As suggested, all acronyms used in our work are explained in Table 1 of the revision.

Reviewer 3 Report

The paper proposed a flexible multi-keyword ranked searchable attribute- based scheme using search trees, which allowed users to combine their fuzzy searching keywords with AND-OR logic gates. Security analysis and experiments were conducted for verification.

Some concerns as follows.

The authors were suggested to give the challenges in introduction and present how to address these challenges  in Section 3.

The authors were suggested to use a table to list the existing works and compare them, in order to highlight the paper contriubtions.

Experiment results are unclear.

The authors should cmpare their work with the existing works of 2021--2023.  

Author Response

Thanks for reviewing and sharing valuable suggestions. Below is our point-to-point reply to the comment:

The authors were suggested to give the challenges in introduction and present how to address these challenges in Section 3.

We list several challenges in Introduction. For example, under traditional PKE, we must copy files many times and encrypt them for each access group respectively. Another challenge is how to fetch data in need rapidly since standard searchable encryption schemes only allow users to fetch files containing all the keywords.

As a response, we present the proposed approach to solve these challenges in the Introduction in short and the detail in chapters 3 and 4.

The authors were suggested to use a table to list the existing works and compare them, in order to highlight the paper contributions.

The comparison can be found in Table 2.

Experiment results are unclear.

Chapter 6 has been revised.

The authors should compare their work with the existing works of 2021--2023.  

There are some related works published recently. For example, “He, W.; Zhang, Y.; Li, Y. Fast, Searchable, Symmetric Encryption Scheme Supporting Ranked Search. Symmetry 2022, 14, 1029. https://doi.org/10.3390/ sym14051029”. A comparison with these works will be done in our future work.

Round 2

Reviewer 2 Report

I recommend this paper for publication.

Author Response

Thank you for the comment in round 1.

Reviewer 3 Report

The paper description was improved much.  There are still some issues to be addressed.

(1) Clearly list what security goals to be achieved and how to address in the paper.

(2) It is unlcear about the difference between two proposed schemes. Is it necessary to give two schemes in the paper? 

(3) check grammar.

(4) Experiment results are unclear.

Author Response

Thanks for reviewing our paper again. Below is our feedback for the round2’s comment:

(1) Clearly list what security goals to be achieved and how to address in the paper.

We have added one more paragraph in the Introduction for the security goal.

(2) It is unlcear about the difference between two proposed schemes. Is it necessary to give two schemes in the paper?

Yes, the example of Fuzzy and Semantic search can be found in Figure 1. Fuzzy search allows input containing some typo and Semantic search extends user input to a large list to help users to find the result. In the E-health use case, both extensions are essential.

(3) check grammar.

We have revised the grammar again.

(4) Experiment results are unclear.

We have updated all the figures in Figure 7 and Figure 8. The resolution is improved.

Round 3

Reviewer 3 Report

There are grammar errors. For example, 'Our approaches provide the most efficient MABKS [17] and MSDVABE 648 [21] schemes.'

The authors  should carefully check the whole paper description.

Author Response

Thanks for the suggestion. We have reviewed the paper again.